# GRADIENT-BASED LEARNING FOR THE $F$-MEASURE AND OTHER PERFORMANCE METRICS

## ABSTRACT

Many important classification performance metrics, e.g. the $F$-measure, are non-differentiable and non-decomposable, and are thus unfriendly to the gradient descent algorithm. Consequently, despite their popularity as evaluation metrics, these metrics are rarely optimized as training objectives in neural network community. In this paper, we propose an empirical utility maximization scheme with provable learning guarantees to address the non-differentiability of these metrics. We then derive a strongly consistent gradient estimator to handle non-decomposability. These innovations enable end-to-end optimization of these performance metrics with the same computational complexity as optimizing a decomposable and differentiable metric, e.g. the cross-entropy loss.

## 1 INTRODUCTION

Different classification performance metrics are capable of revealing different aspects of a classifier's behavior. For example, the $F$-measure (Van Rijsbergen (1974)), compared to performance metrics such as accuracy, is better at evaluating a classifier's performance when it encounters a sample belonging to a class that occurs with low frequency. Ideally, we can acquire a classifier with very tailored behavior by optimizing the classifier with respect to a carefully chosen performance metric. Unfortunately, many performance metrics, e.g. the $F$-measure, are non-differentiable and non-decomposable, which renders it very difficult to optimize neural network classifiers with these metrics as training objective.

In this paper, we propose a method that enables gradient-based learning for these performance metrics. Our contributions are the following:

- We propose a learning algorithm based on empirical utility maximization for a class of performance metrics and prove its generalization and consistency.
- We propose a strongly consistent gradient estimator that enables efficient gradient-based maximization of empirical utility.
- We demonstrate experimentally that the binary $F_1$ score of neural network classifiers can be efficiently optimized on datasets of decent scale and complexity.

We organize this paper as the following. In Section 2, we will sketch our method for the binary $F_1$ score to provide an overview. In Section 3, we will present our method in its general form. We review related work in Section 4 and provide experiment results in Section 5.

## 2 GRADIENT-BASED LEARNING FOR THE BINARY $F_1$ SCORE

### 2.1 PROBABILISTIC CLASSIFIER

Given a feature vector $\boldsymbol{x} \in \mathcal{X} \subset \mathbb{R}^N$, a probabilistic classifier $h$ first infers a posterior $p(\cdot|\boldsymbol{x})$ over a discrete output space $\mathcal{Y}$ and then samples its output from the posterior, i.e. $h(\boldsymbol{x}) \sim p(\cdot|\boldsymbol{x})$. In practice, $p(\cdot|\boldsymbol{x})$ is typically the output of a neural network with softmax layer on its top.When the posterior is parameterized, e.g. being implemented as a neural network, we denote it as $p_\theta(\cdot|\boldsymbol{x})$ and the corresponding probabilistic classifier as $h_\theta$.

Given a posterior $p(\cdot|\boldsymbol{x})$, a deterministic classifier can result from the inference rule $h(\boldsymbol{x}) = \text{argmax}_{y \in \mathcal{Y}} p(y|\boldsymbol{x})$. The difference between probabilistic and deterministic inference rules is negligible when the posterior is very concentrated. Although deterministic classifiers are more popular in the literature, in this paper we only consider probabilistic classifiers and leave it as future work to investigate the case where a probabilistic classifier is replaced by a deterministic one.

## 2.2 $F$-MEASURE

Consider the case of binary classification, where $\mathcal{Y} = \{0, 1\}$ with $1$ and $0$ respectively corresponding to the positive and negative class. Given a dataset $D = \{(\boldsymbol{x}_1, y_1), ..., (\boldsymbol{x}_n, y_n)\}$ consisting of $n$ i.i.d. pairs of feature vector and ground truth, let $\hat{y}_i$ denote the label predicted by a classifier $h$ given $\boldsymbol{x}_i$ (not necessarily deterministically). Let $\hat{\boldsymbol{y}} = (\hat{y}_1, ..., \hat{y}_n)$ and $\boldsymbol{y} = (y_1, ..., y_n)$. Then the true-positive, false-positive, false-negative and true-negative rate corresponding to $\hat{\boldsymbol{y}}$ and $\boldsymbol{y}$ are defined as

$$\text{tp}(\hat{\boldsymbol{y}}, \boldsymbol{y}) := \frac{1}{n} \sum_{i=1}^{n} \mathbb{I}(\hat{y}_i = 1 \wedge y_i = 1) \qquad \text{fp}(\hat{\boldsymbol{y}}, \boldsymbol{y}) := \frac{1}{n} \sum_{i=1}^{n} \mathbb{I}(\hat{y}_i = 1 \wedge y_i = 0)$$

$$\text{fn}(\hat{\boldsymbol{y}}, \boldsymbol{y}) := \frac{1}{n} \sum_{i=1}^{n} \mathbb{I}(\hat{y}_i = 0 \wedge y_i = 1) \qquad \text{tn}(\hat{\boldsymbol{y}}, \boldsymbol{y}) := \frac{1}{n} \sum_{i=1}^{n} \mathbb{I}(\hat{y}_i = 0 \wedge y_i = 0)$$

where $\mathbb{I}$ denotes indicator function. The precision and recall are defined as

$$\text{precision}(\hat{\boldsymbol{y}}, \boldsymbol{y}) = \frac{\text{tp}(\hat{\boldsymbol{y}}, \boldsymbol{y})}{\text{tp}(\hat{\boldsymbol{y}}, \boldsymbol{y}) + \text{fp}(\hat{\boldsymbol{y}}, \boldsymbol{y})} \qquad \text{recall}(\hat{\boldsymbol{y}}, \boldsymbol{y}) = \frac{\text{tp}(\hat{\boldsymbol{y}}, \boldsymbol{y})}{p_D^+} \qquad (1)$$

where $p_D^+ := \frac{1}{n} \sum_{i=1}^{n} \mathbb{I}(y_i = 1)$ denotes the proportion of samples in $D$ that belong to positive class. The binary $F$-measure is defined as (Van Rijsbergen (1974)):

$$F_\beta(\hat{\boldsymbol{y}}, \boldsymbol{y}) = (1 + \beta^2) \cdot \frac{\text{precision}(\hat{\boldsymbol{y}}, \boldsymbol{y}) \cdot \text{recall}(\hat{\boldsymbol{y}}, \boldsymbol{y})}{\beta \cdot \text{precision}(\hat{\boldsymbol{y}}, \boldsymbol{y}) + \text{recall}(\hat{\boldsymbol{y}}, \boldsymbol{y})} \qquad \beta > 0 \qquad (2)$$

or equivalently,

$$F_\beta(\hat{\boldsymbol{y}}, \boldsymbol{y}) = (1 + \beta^2) \cdot \frac{p_D^+ - \text{fn}(\hat{\boldsymbol{y}}, \boldsymbol{y})}{(1 + \beta^2)p_D^+ - \text{fn}(\hat{\boldsymbol{y}}, \boldsymbol{y}) + \text{fp}(\hat{\boldsymbol{y}}, \boldsymbol{y})} \qquad \beta > 0 \qquad (3)$$

which is more convenient for our purpose.

We will refer to $F_\beta(\hat{\boldsymbol{y}}, \boldsymbol{y})$ as the *data-dependent* binary $F_\beta$-measure because it is evaluated on a specific set of data with pairs of ground truth and prediction vectors. $F_\beta$ is non-differentiable because it is a composition of indicator functions. Nor does it decompose over samples in $D$. More precisely, we are not aware of any function $f_\beta$ that only depends on per sample ground-truth and prediction such that

$$F_\beta(\hat{\boldsymbol{y}}, \boldsymbol{y}) = \frac{1}{n} \sum_{i=1}^{n} f_\beta(\hat{y}_i, y_i)$$

In the following we propose an empirical utility maximization scheme for optimizing the $F_\beta$-measure of probabilistic classifiers. For ease of exposition, in this section we focus on the binary $F_1$-measure, a.k.a. the binary $F_1$ score. In Section 3, we will extend the method presented in this section to a family of non-decomposable and non-differentiable performance metrics, including $F_\beta$-measure for multi-class classification.

## 2.3 GRADIENT-BASED LEARNING FOR THE BINARY $F_1$ SCORE

We consider a parameterized binary probabilistic classifier $h_\theta$. By linearity of expectation and the i.i.d. assumption,

$$\mathbb{E}_{\hat{\boldsymbol{y}}, \boldsymbol{y}}[\text{fp}(\hat{\boldsymbol{y}}, \boldsymbol{y})] = \mathbb{P}(\hat{\text{y}} = 1 \wedge \text{y} = 0)$$

where the expectation is taken over all datasets with a fixed size $n$ and all possible predictions of $h_\theta$. Similarly, $\mathbb{E}_{\hat{\boldsymbol{y}}, \boldsymbol{y}}[\text{fn}(\hat{\boldsymbol{y}}, \boldsymbol{y})] = \mathbb{P}(\hat{\text{y}} = 0 \wedge \text{y} = 1)$. Let $\overline{\text{fn}}(h_\theta) := \mathbb{E}_{\hat{\boldsymbol{y}}, \boldsymbol{y}}[\text{fn}(\hat{\boldsymbol{y}}, \boldsymbol{y})]$ and $\overline{\text{fp}}(h_\theta) := \mathbb{E}_{\hat{\boldsymbol{y}}, \boldsymbol{y}}[\text{fp}(\hat{\boldsymbol{y}}, \boldsymbol{y})]$. It follows from the law of large number that

$$\lim_{|D| \to \infty} \text{fn}(\hat{\boldsymbol{y}}, \boldsymbol{y}) = \overline{\text{fn}}(h_\theta) \qquad \lim_{|D| \to \infty} \text{fp}(\hat{\boldsymbol{y}}, \boldsymbol{y}) = \overline{\text{fp}}(h_\theta)$$

with probability 1, where $|D|$ denotes the size of dataset $D$. Thus on sufficiently large datasets,

$$\text{fn}(\hat{\boldsymbol{y}}, \boldsymbol{y}) \approx \overline{\text{fn}}(h_\theta) \qquad \text{fp}(\hat{\boldsymbol{y}}, \boldsymbol{y}) \approx \overline{\text{fp}}(h_\theta)$$

With these approximate identities, we have the following approximation of $F_1(\hat{\boldsymbol{y}}, \boldsymbol{y})$:

$$F_1(\hat{\boldsymbol{y}}, \boldsymbol{y}) = 2 \cdot \frac{p_D^+ - \text{fn}(\hat{\boldsymbol{y}}, \boldsymbol{y})}{2p_D^+ - \text{fn}(\hat{\boldsymbol{y}}, \boldsymbol{y}) + \text{fp}(\hat{\boldsymbol{y}}, \boldsymbol{y})} \approx 2 \cdot \frac{p_D^+ - \overline{\text{fn}}(h_\theta)}{2p_D^+ - \overline{\text{fn}}(h_\theta) + \overline{\text{fp}}(h_\theta)} := \bar{F}_1(h_\theta) \qquad (4)$$

which implies that the $F_1$ score of *any* predictions of $h_\theta$ on *any* sufficiently large dataset is close to $\bar{F}_1(h_\theta)$. We call $\bar{F}_1(h_\theta)$ the *expected* utility of the $F_1$ score and will state the precise meaning of $F_1(\hat{\boldsymbol{y}}, \boldsymbol{y}) \approx \bar{F}_1(h_\theta)$ in Section 3. The key point is that we can optimize $\bar{F}_1(h_\theta)$ instead of $F_1(\hat{\boldsymbol{y}}, \boldsymbol{y})$ if we are interested in the $F_1$ score of $h_\theta$ on sufficiently large datasets. However, $\overline{\text{fn}}(h_\theta)$ and $\overline{\text{fp}}(h_\theta)$ are unknown because they are expectations taken over data distribution (and the classifier's posterior). Consequently, we have to estimate $\overline{\text{fn}}(h_\theta)$ and $\overline{\text{fp}}(h_\theta)$ by sampling from data distribution in order to estimate $\bar{F}_1(h_\theta)$, as the following.

Let $p^+ := \mathbb{P}(\text{y} = 1)$ denote the probability that a positive sample occurs, which can be estimated by the frequency of positive samples in a training set $D$. Let $n^+ := \sum_{(\boldsymbol{x}, y) \in D} \mathbb{I}(y = 1)$ denote the number of positive samples in the training set. Assume that the data distribution admits a density function (i.e. the data distribution is absolutely continuous w.r.t. the Lebesgue measure), and denote its density function by $p$. We have the following unbiased estimator of $\overline{\text{fn}}(h_\theta)$:

$$\begin{aligned}
\overline{\text{fn}}(h_\theta) &= \mathbb{P}\left(\hat{\text{y}} = 0 \wedge \text{y} = 1\right) \\
&= \int_{\mathcal{X}} \mathbb{P}(h_\theta(\boldsymbol{x}) = 0) p(\boldsymbol{x}, 1) \, \mathrm{d}\boldsymbol{x} \\
&= \int_{\mathcal{X}} p_\theta(0|\boldsymbol{x}) p(\boldsymbol{x}|1) p^+ \, \mathrm{d}\boldsymbol{x} \\
&= p^+ \int_{\mathcal{X}} p_\theta(0|\boldsymbol{x}) p(\boldsymbol{x}|1) \, \mathrm{d}\boldsymbol{x} \\
&= p^+ \mathbb{E}_{\boldsymbol{x} \sim p(\cdot|1)}[p_\theta(0|\boldsymbol{x})] \\
&\approx \frac{p^+}{n^+} \sum_{i=1}^{n^+} p_\theta(0|\boldsymbol{x}_i^+) := \widehat{\text{fn}}_D(h_\theta)
\end{aligned} \qquad (5)$$

where $\boldsymbol{x}_1^+, ..., \boldsymbol{x}_{n^+}^+$ are the feature vectors of samples belonging to the positive class in trainingset $D$. Similarly,

$$\overline{\text{fp}}(h_\theta) = \mathbb{P}\left(\hat{\text{y}} = 1 \wedge \text{y} = 0\right) \approx \frac{p^-}{n^-} \sum_{i=1}^{n^-} p_\theta(1|\boldsymbol{x}_i^-) := \widehat{\text{fp}}_D(h_\theta)$$

where $p^- := \mathbb{P}(\text{y} = 0)$, $n^- := \sum_{(\boldsymbol{x}, y) \in D} \mathbb{I}(y = 0)$, and $\boldsymbol{x}_1^-, ..., \boldsymbol{x}_n^-$ are the feature vectors of samples in $D$ belonging to the negative class. Thus $\bar{F}_1(h_\theta)$ can be estimated as the following:

$$\bar{F}_1(h_\theta) = 2 \cdot \frac{p^+ - \overline{\text{fn}}(h_\theta)}{2p^+ - \overline{\text{fn}}(h_\theta) + \overline{\text{fp}}(h_\theta)} \approx 2 \cdot \frac{p^+ - \widehat{\text{fn}}_D(h_\theta)}{2p^+ - \widehat{\text{fn}}_D(h_\theta) + \widehat{\text{fp}}_D(h_\theta)} := \hat{F}_D(h_\theta) \qquad (6)$$

We will state the precise meaning of $\bar{F}_1(h_\theta) \approx \hat{F}_D(h_\theta)$ in Section 3. Interestingly, although $\text{fn}(\hat{\boldsymbol{y}}, \boldsymbol{y})$ and $\text{fp}(\hat{\boldsymbol{y}}, \boldsymbol{y})$ are not differentiable, the estimators of their expectations, $\widehat{\text{fn}}_D(h_\theta)$ and $\widehat{\text{fp}}_D(h_\theta)$, are differentiable w.r.t. $\theta$ if $p_\theta$ is differentiable. Because $\hat{F}_D(h_\theta)$ is differentiable w.r.t. $\widehat{\text{fn}}(h_\theta)$ and $\widehat{\text{fp}}(h_\theta)$, $\nabla_\theta \hat{F}_D(h_\theta)$ can be computed by chain rule. Consequently, gradient descent can be applied to optimize $\hat{F}_D(\theta)$.

We call $\hat{F}_D(\theta)$ the *empirical utility* of the expected utility $\bar{F}_1(h_\theta)$. They correspond to empirical and expected risk in the classical empirical risk minimization principle of statistical learning theory (Vapnik (1992)). We use the term "empirical utility maximization" because we would like to maximize, instead of minimize these performance metrics. There are two fundamental questions for every empirical risk minimization style learning algorithm, as the number of samples increases:

- Generalization. Given $h_\theta$, does $\hat{F}_D(h_\theta) \to \bar{F}_1(h_\theta)$ as $|D| \to \infty$?
- Consistency. Does $\text{argmax}_\theta \hat{F}_D(h_\theta) \to \text{argmax}_\theta \bar{F}_1(h_\theta)$ as $|D| \to \infty$?

We will address these two questions at the end of Section 3. For the moment let us consider a practical issue: how to maximize empirical utility $\bar{F}_D(h_\theta)$ efficiently with gradient descent?

## 2.4 GRADIENT ESTIMATOR

In order for the approximation in Eq. 6 to be accurate, $|D|$ has to be sufficiently large. In order to optimize $\hat{F}_D(h_\theta)$ efficiently via minibatch gradient descent, $\nabla_\theta \hat{F}_D(h_\theta)$ has to be estimated by $\nabla_\theta \hat{F}_B(h_\theta)$, where $B \subset D$ is a mini-batch, such that $\mathbb{E}_B[\nabla_\theta \hat{F}_B(h_\theta)] = \nabla_\theta \hat{F}_D(h_\theta)$. Suppose $\hat{F}_D(h_\theta)$ is decomposable, i.e. there is a per-sample loss function $\hat{f}$ such that $\hat{F}_D(h_\theta) = \frac{1}{|D|} \sum_{(\boldsymbol{x},y) \in D} \hat{f}(p_\theta(\boldsymbol{x}), y)$, then it simply follows from linearity of differentiation and expectation that the requirement is satisfied. However, as $\hat{F}_D(h_\theta)$ is non-decomposable, it becomes unlikely that $\nabla_\theta \hat{F}_B(h_\theta)$ is an unbiased estimator of $\nabla_\theta \hat{F}_D(h_\theta)$. Fortunately, as a consequence of Theorem 1, $\nabla_\theta \hat{F}_B(h_\theta)$ is a strongly consistent estimator of $\nabla_\theta \hat{F}_D(h_\theta)$ when $|D|$ is sufficiently large. More precisely,

$$\mathbb{P}\left( \lim_{|B| \to \infty} \nabla_\theta \hat{F}_B(h_\theta) = \lim_{|D| \to \infty} \nabla_\theta \hat{F}_D(h_\theta) \right) = 1 \tag{7}$$

Thus $\nabla_\theta \hat{F}_B(h_\theta)$ is almost as good as an unbiased estimator. More interestingly, the error incurred by estimating $\nabla_\theta \hat{F}_D(h_\theta)$ with $\nabla_\theta \hat{F}_B(h_\theta)$ can be further controlled. In the following we omit the dependence on $h_\theta$ for brevity. Let $\boldsymbol{\phi}_D := (\widehat{\text{fn}}_D(h_\theta), \widehat{\text{fp}}_D(h_\theta)) \in \mathbb{R}^2$ and $\boldsymbol{\phi}_B := (\widehat{\text{fn}}_B(h_\theta), \widehat{\text{fp}}_B(h_\theta)) \in \mathbb{R}^2$. Let $\hat{\boldsymbol{J}}_D$ and $\hat{\boldsymbol{J}}_B$ denotes the Jacobian of $\hat{\boldsymbol{\phi}}_D$ and $\hat{\boldsymbol{\phi}}_B$ w.r.t. $\theta$. Let $|\cdot|$ denote a vector norm and $||\cdot||$ denote the matrix norm induced by it. By chain rule,

$$\begin{aligned}
\nabla_\theta \hat{F}_B &= \nabla_{\boldsymbol{\phi}_B} \hat{F}_B \hat{\boldsymbol{J}}_B \\
&= \left( \nabla_{\boldsymbol{\phi}_D} \hat{F}_D + \boldsymbol{\epsilon} \right) \left( \hat{\boldsymbol{J}}_D + \boldsymbol{\mathcal{E}} \right) \\
&= \nabla_{\boldsymbol{\phi}_D} \hat{F}_D \hat{\boldsymbol{J}}_D + \nabla_{\boldsymbol{\phi}_D} \hat{F}_D \boldsymbol{\mathcal{E}} + \boldsymbol{\epsilon} \hat{\boldsymbol{J}}_D + \boldsymbol{\epsilon} \boldsymbol{\mathcal{E}}
\end{aligned} \tag{8}$$

where $\nabla_{\boldsymbol{\phi}_D} \hat{F}_D \cdot \hat{\boldsymbol{J}}_D$ is the true gradient and $\boldsymbol{\epsilon} \cdot \boldsymbol{\mathcal{E}}$ is negligible. The error $\boldsymbol{\mathcal{E}} = \hat{\boldsymbol{J}}_B(h_\theta) - \hat{\boldsymbol{J}}_D(h_\theta)$ is intrinsic in the sense that it results immediately from estimating $\nabla_\theta \widehat{\text{fn}}_D(h_\theta)$ and $\nabla_\theta \widehat{\text{fp}}_D(h_\theta)$ with $\nabla_\theta \widehat{\text{fn}}_B(h_\theta)$ and $\nabla_\theta \widehat{\text{fp}}_B(h_\theta)$ and it is always present in mini-batch gradient descent because $\mathbb{E}[\nabla_\theta \widehat{\text{fn}}_B(h_\theta)] = \nabla_\theta \widehat{\text{fn}}_D(h_\theta)$ and $\mathbb{E}[\nabla_\theta \widehat{\text{fp}}_B(h_\theta)] = \nabla_\theta \widehat{\text{fp}}_D(h_\theta)$. However, we can control the error $\boldsymbol{\epsilon} \cdot \hat{\boldsymbol{J}}_D$ because $|\boldsymbol{\epsilon} \cdot \hat{\boldsymbol{J}}| \le ||\hat{\boldsymbol{J}}|| \cdot |\boldsymbol{\epsilon}|$ and we can control $||\hat{\boldsymbol{J}}||$ by limiting $|\theta|$ and the norm of intermediate activations when $p_\theta$ is a neural network (He et al. (2015)). Despite these technicalities, the trick is very easy to implement: batch normalization (Ioffe & Szegedy (2015)) and weight decay will suffice. These are summarized in Algorithm 1.

---

**Algorithm 1** Gradient-based learning for the binary $F_1$ score

---

**Require:** classifier $h_\theta$, batch size $b$, dataset $D$, learning rate $\alpha$, weight decay strength $\lambda$
  $p^+ \leftarrow \frac{1}{|D|} \sum_{(\boldsymbol{x},y) \in D} \mathbb{I}(y = 1)$
  $p^- \leftarrow \frac{1}{|D|} \sum_{(\boldsymbol{x},y) \in D} \mathbb{I}(y = 0)$
  **while** terminating criterion not satisfied **do**
    Sample $B_+ = \{(\boldsymbol{x}_1^+, 1), ..., (\boldsymbol{x}_b^+, 1)\}$ from $D$
    Sample $B_- = \{(\boldsymbol{x}_1^-, 0), ..., (\boldsymbol{x}_b^-, 0)\}$ from $D$
    $\text{fn} \leftarrow \frac{p^+}{b} \sum_{i=1}^b p_\theta(0|\boldsymbol{x}_1^+)$
    $\text{fp} \leftarrow \frac{p^-}{b} \sum_{i=1}^b p_\theta(1|\boldsymbol{x}_1^-)$
    $\delta \leftarrow \nabla_\theta \left( F_1(\text{fn}, \text{fp}) - \lambda \cdot |\theta| \right)$
    $\theta \leftarrow \theta + \alpha \cdot \delta$
  **end while**

---

## 3 GRADIENT-BASED LEARNING FOR A CLASS OF PERFORMANCE METRICS

The binary $F$-measure in fact belongs to a class of performance metrics that are well behaved functions of the confusion matrix. In this section, we propose a gradient-based learning algorithm that extends the approach illustrated in previous section to this class of performance metrics. We state theorems concerning the generalization and consistency of the proposed algorithm as well. We defer all proofs to appendix. We begin with a specification of this class of performance metrics, which relies on the definition of confusion matrix:

**Definition 1.** *Given a dataset $D = \{(\boldsymbol{x}_1, y_1), ..., (\boldsymbol{x}_n, y_n)\}$, let $\boldsymbol{y} = (y_1, ..., y_n)$ denote the vector of ground truth and $\hat{\boldsymbol{y}} = (\hat{y}_1, ..., \hat{y}_n)$ denote a vector of classifier predictions. Then the corresponding* data-dependent *confusion matrix $C(\hat{\boldsymbol{y}}, \boldsymbol{y})$ is defined as*

$$\left(C(\hat{\boldsymbol{y}}, \boldsymbol{y})\right)_{ij} := \frac{1}{n} \sum_{k=1}^{n} \mathbb{I}\left(\hat{y}_k = i \wedge y_k = j\right) \qquad 0 \le i, j \le |\mathcal{Y}| - 1$$

In the case of binary classification,

$$C(\hat{\boldsymbol{y}}, \boldsymbol{y}) = \begin{bmatrix} \text{tn}(\hat{\boldsymbol{y}}, \boldsymbol{y}) & \text{fn}(\hat{\boldsymbol{y}}, \boldsymbol{y}) \\ \text{fp}(\hat{\boldsymbol{y}}, \boldsymbol{y}) & \text{tp}(\hat{\boldsymbol{y}}, \boldsymbol{y}) \end{bmatrix}$$

Let $\mathbb{P}$ be a probability measure induced by data distribution and a probabilistic classifier $h_\theta$ over $\mathcal{X} \times \mathcal{Y} \times \mathcal{Y}$, i.e. triples of feature vector, ground truth and classifier prediction. Let $\bar{C}(h_\theta)$ be the entry-wise expectation of $C(\hat{\boldsymbol{y}}, \boldsymbol{y})$ over $\mathbb{P}$, referred to as the *expected* confusion matrix. Formally,

$$\left(\bar{C}(h_\theta)\right)_{ij} := \mathbb{E}\left[\mathbb{I}(h_\theta(\mathbf{x}) = i \wedge \mathbf{y} = j)\right] = \mathbb{P}(h_\theta(\mathbf{x}) = i \wedge \mathbf{y} = j) \qquad 0 \le i, j \le |\mathcal{Y}| - 1$$

As in Section 2, given a training set $D$, we have the following unbiased estimator of $\bar{C}(h_\theta)$, referred to as the *empirical* confusion matrix:

$$\left(\hat{C}_D(h_\theta)\right)_{ij} = \frac{p_j}{n_j} \sum_{k=1}^{n_j} p_\theta(i|\boldsymbol{x}_k^j) \qquad 0 \le i, j \le |\mathcal{Y}| - 1$$

where $n_j := \sum_{(\boldsymbol{x}, y) \in D} \mathbb{I}(y = j)$, $p_j := \frac{n_j}{|D|}$, and $\boldsymbol{x}_1^j, ..., \boldsymbol{x}_{n_j}^j$ are the feature vectors of samples that belong to the $j$-th class. Almost sure convergence follows from the law of large number:

$$\mathbb{P}\left(\lim_{|D| \to \infty} \left(C(\hat{\boldsymbol{y}}, \boldsymbol{y})\right)_{ij} = \left(\bar{C}(h_\theta)\right)_{ij}\right) = 1 \qquad \mathbb{P}\left(\lim_{|D| \to \infty} \left(\hat{C}_D(h_\theta)\right)_{ij} = \left(\bar{C}(h_\theta)\right)_{ij}\right) = 1 \quad (9)$$

The confusion matrix is well-defined for both single-label and multi-label classification (although these two settings impose different constraints on its entries). Many performance metrics are functions of the confusion matrix. For example, the accuracy of $h_\theta$ is $\sum_{i=1}^{|\mathcal{Y}|} \bar{C}_{ii}(h_\theta)$. The $F_\beta$ measure for multi-class classification can be defined in term of entries of the confusion matrix as the following. We first define for every class the data-dependent false positive and false negative rate as

$$\text{fp}_i = \sum_{j \ne i} C_{ij} \qquad \text{fn}_i = \sum_{j \ne i} C_{ji} \qquad i = 1, ..., |\mathcal{Y}|$$

where we omit the dependence on $\hat{\boldsymbol{y}}$ and $\boldsymbol{y}$ for brevity. The data-dependent macro and micro $F$-measure (Parambath et al. (2014)) are defined in term of $\text{fp}_i$ and $\text{fn}_i$ as

$$F_\beta^{\text{macro}} = \frac{1 + \beta^2}{|\mathcal{Y}|} \sum_{i=1}^{|\mathcal{Y}|} \frac{p_i - \text{fn}_i}{(1 + \beta^2)p_i - \text{fn}_i + \text{fp}_i} \qquad \beta > 0$$

$$F_\beta^{\text{micro}} = (1 + \beta^2) \cdot \frac{\sum_{i=1}^{|\mathcal{Y}|} p_i - \sum_{i=1}^{|\mathcal{Y}|} \text{fn}_i}{(1 + \beta^2) \sum_{i=1}^{|\mathcal{Y}|} p_i - \sum_{i=1}^{|\mathcal{Y}|} \text{fn}_i + \sum_{i=1}^{|\mathcal{Y}|} \text{fp}_i} \qquad \beta > 0$$

Replacing $C(\hat{\boldsymbol{y}}, \boldsymbol{y})$ by $\bar{C}(h_\theta)$ and $\hat{C}(h_\theta)$ in these definitions will respectively result in the expected and empirical $F$-measure.

We now specify the class of performance metrics that we are interested in, namely the class of well-behaved performance metrics. In the following, we will consider $C(\hat{\boldsymbol{y}}, \boldsymbol{y})$, $\bar{C}(h_\theta)$ and $\hat{C}_D(h_\theta)$ as vectors of dimension $|\mathcal{Y}| \times |\mathcal{Y}|$ and identify a performance metric with a function that maps $|\mathcal{Y}| \times |\mathcal{Y}|$-dimensional vectors to real values.

**Definition 2.** *We say that a performance metric $F : K \mapsto \mathbb{R}$, where $K$ is a compact subset of $\mathbb{R}^{|\mathcal{Y}| \times |\mathcal{Y}|}$, is well-behaved if $F$ is continuously differentiable on $K$.*

Please refer to appendix for a non-exhaustive list of well-behaved performance metrics. Importantly, the binary $F_1$ score and the macro and micro $F$-measure are well-behaved performance metrics (proof deferred to appendix).

Given a well behaved performance metric $F$, its corresponding data-dependent, expected and empirical utility are respectively defined as $F(C(\hat{\boldsymbol{y}}, \boldsymbol{y}))$, $F(\bar{C}(h_\theta))$ and $F(\hat{C}(h_\theta))$. The following theorem establishes asymptotic equivalence between these three kinds of utilities.

**Theorem 1.** *If $F$ is a well-behaved performance metric and $C_D$ is a strongly consistent estimator of $C$, i.e.*

$$\mathbb{P}\left(\lim_{|D| \to \infty} (C_D)_{ij} = C_{ij}\right) = 1 \qquad 0 \le i, j \le |\mathcal{Y}| - 1$$

*then $F(C_D)$ is a strongly consistent estimator of $F(C)$, i.e.*

$$\mathbb{P}\left(\lim_{|D| \to \infty} F(C_D) = F(C)\right) = 1$$

As a consequence of this theorem, it follows from Eq. 9 that

$$\mathbb{P}\left(\lim_{|D| \to \infty} F(C(\hat{\boldsymbol{y}}, \boldsymbol{y})) = F(\bar{C}(h_\theta))\right) = 1 \qquad \mathbb{P}\left(\lim_{|D| \to \infty} F(\hat{C}_D(h_\theta)) = F(\bar{C}(h_\theta))\right) = 1$$

i.e. both $F(C(\hat{\boldsymbol{y}}, \boldsymbol{y}))$ and $F(\hat{C}(h_\theta))$ are strongly consistent estimators of (converge w.p. 1 to) $F(\bar{C}(h_\theta))$. As a special case,

$$\mathbb{P}\left(\lim_{|D| \to \infty} F_1(\hat{\boldsymbol{y}}, \boldsymbol{y}) = \bar{F}_1(h_\theta)\right) = 1 \qquad \mathbb{P}\left(\lim_{|D| \to \infty} \hat{F}_1(h_\theta) = \bar{F}_1(h_\theta)\right) = 1$$

which justifies Eq. 4 and Eq. 6 when the dataset of interest is sufficiently large. Next we consider the issue of gradient estimation in this general setting.

**Theorem 2.** *If $F$ is a well behaved performance metric, then $\nabla_\theta F(\hat{C}_B(h_\theta))$ is a strongly consistent estimator of $\nabla_\theta F(\bar{C}(h_\theta))$, where $B \subset D$ is a mini-batch. More precisely,*

$$\mathbb{P}\left(\lim_{|B| \to \infty} \nabla_\theta F\left(\hat{C}_B(h_\theta)\right) = \nabla_\theta F\left(\bar{C}(h_\theta)\right)\right) = 1$$

As proved in Chen & Luss (2018), many convergence guarantees for stochastic gradient descent with unbiased gradient estimators holds for stochastic gradient descent with strongly consistent gradient estimators with probability 1. As illustrated in Eq. 8, batch normalization and weight decay can help control the noise of estimator. Please refer to Algorithm 2 for the resultant algorithm.

Finally, we state two theorems concerning the generalization and consistency of Algorithm 2. Rates of convergence are omitted for brevity.

**Theorem 3** (Generalization). *For a well behaved performance metric $F$, for all $\epsilon > 0$,*

$$\lim_{|D| \to \infty} \mathbb{P}\left(\left|F(\hat{C}_D(h_\theta)) - F(\bar{C}(h_\theta))\right| < \epsilon\right) = 1$$

**Theorem 4** (Consistency). *For a well behaved performance metric $F$, with appropriate constraints on the capacity of parametric model $p_\theta$ (see the proof for details), we have that for all $\epsilon > 0$,*

$$\lim_{|D| \to \infty} \mathbb{P}\left(\left|\arg\max_\theta F\left(\hat{C}_D(h_\theta)\right) - \arg\max_\theta F\left(\bar{C}(h_\theta)\right)\right| < \epsilon\right) = 1$$

*i.e. Algorithm 2 is consistent.*

---

**Algorithm 2** Gradient-based learning for well behaved performance metrics

---

**Require:** batch size $b$, classifier $h_\theta$, dataset $D$, learning rate $\alpha$, weight-decay strength $\lambda$, and well-behaved metric $F$

    **for** $i = 1, ..., |\mathcal{Y}|$ **do**

        $p_i \leftarrow \frac{1}{|D|} \sum_{(\boldsymbol{x},y) \in D} \mathbb{I}(y = i)$

    **end for**

    **while** terminating criterion not satisfied **do**

        **for** $i = 1, ..., |\mathcal{Y}|$ **do**

            Sample $B_i = \{(\boldsymbol{x}_1^i, i), ..., (\boldsymbol{x}_b^i, i)\}$ from $D$

            Compute $p_\theta(\cdot|\boldsymbol{x}_1^i), ..., p_\theta(\cdot|\boldsymbol{x}_b^i)$

        **end for**

        **for** $i = 1, ..., |\mathcal{Y}|$ **do**

            **for** $j = 1, ..., |\mathcal{Y}|$ **do**

                $C_{ij} \leftarrow \frac{p_j}{b} \sum_{k=1}^{b} p_\theta(i|\boldsymbol{x}_k^j)$

            **end for**

        **end for**

        $\delta \leftarrow \nabla_\theta \left( F(C) - \lambda|\theta| \right)$

        $\theta \leftarrow \theta + \alpha \cdot \delta$

    **end while**

---

## 4 RELATED WORK

The optimization of non-decomposable and non-differentiable performance metrics, especially $F$-measure, has been extensively studied. The heuristic algorithm considered in Jansche (2005) and Pastor-Pellicer et al. (2013) is essentially Algorithm 1 without techniques that stabilize gradient estimation. However, Jansche (2005) and Pastor-Pellicer et al. (2013) are not very well motivated theoretically and provide little mathematical insight into the heuristic. Also, as shown in Section 5, applying this heuristic algorithm without stabilization techniques can easily result in non-convergent models, even for a three-layer fully-connected network.

Another series of papers (Joachims (2005), Kar et al. (2014) and Narasimhan et al. (2015)) study optimizing differentiable lower bounds of various non-decomposable and non-differentiable binary classification metrics for linear classifiers. Despite proved learning guarantees for linear classifier, these lower bound methods are not very promising when applied to neural networks, as reported in Sanyal et al. (2018).

Thresholding is a computationally economical method if we only consider binary classification. Koyejo et al. (2014) proves that the optimal classifier with respect to a family of binary classification metrics, including the $F$-measure, is appropriately thresholded Bayes classifier. Given an approximation of Bayes classifier, we can approach the optimal threshold via grid search. However, it remains unknown how to generalize thresholding to multi-class classification. More importantly, for binary classification, when training set is extremely imbalanced, it can be very difficult to train a classifier that approximates Bayes classifier very well.

The computational cost of aforementioned methods roughly equals that of training classifiers with standard classification losses such as cross-entropy. As proved in Parambath et al. (2014) and Koyejo et al. (2014), optimization of many performance metrics, including $F$-measure, can be reduced to weighted classification. Unfortunately, the optimal weight is in general unknown and has to be approximated by an expensive grid search (see Section 5). Despite its computational cost, unlike thresholding, this method can perform reasonably well even when a training set is extremely balanced. Eban et al. (2016) proposes a similar method that performs well for neural networks coupled with the AUCPR metric.

Regarding theory, the equivalence between the data-dependent and expected utility of the $F$-measure was first proved in Nan et al. (2012) and later generalized in Dembczyński et al. (2017) to $p$-Lipschitz binary classification performance metrics.

Table 1: Dataset statistics and results

| DATASET | # FEATS | # SAMPS | % POS | $F_1$ GS | $F_1$ EUM |
|---------|---------|---------|-------|----------|-----------|
| Adult | 108 | 48,842 | 23.93 | **0.701** | 0.689 |
| CIFAR10 | 3072 | 60,000 | 10.00 | 0.630 | **0.635** |
| Letter | 16 | 20,000 | 3.92 | **0.990** | 0.975 |
| Covtype | 54 | 581,012 | 1.63 | 0.691 | **0.725** |
| CIFAR100 | 3072 | 60,000 | 1.00 | 0.350 | **0.392** |
| KDDCup08 | 117 | 102,294 | 0.61 | 0.543 | **0.556** |

## 5 EXPERIMENTS

We evaluate Algorithm 1 on the following datasets: Letter[1], Adult[2], Covertype[3], KDDCup08[4], CIFAR10 and CIFAR100 (Krizhevsky (2009)). The performance metric to optimize is the binary $F_1$ score. The purpose of this experiment is to demonstrate that Algorithm 1 can match the performance of a provably optimal, yet considerably more expensive algorithm that optimizes the $F_1$ score. We use a three-layer fully-connected network in our experiments, with batch normalization enabled. The statistics of these datasets are summarized in Table 1 (number of features, number of samples, percentage of positive samples). For multi-class datasets (Letter, Covertype, CIFAR10 and CIFAR100), we designate one class as the positive class and leave the rest as the negative class. We compare Algorithm 1 with the following baseline (Parambath et al. (2014) and Koyejo et al. (2014)):

$$\theta^* \leftarrow \underset{\theta, \lambda \in (0,1)}{\arg\max} \frac{1}{|D|} \sum_{(\boldsymbol{x}, y) \in D} l(p_\theta(\boldsymbol{x}), y) \left( \lambda \mathbb{I}(y=0) + (1-\lambda)\mathbb{I}(y=1) \right)$$

where $l$ denotes the cross-entropy loss. We let $\lambda = 0.1, 0.2, 0.3, ..., 0.9$, and apply gradient descent to optimize $\theta$ for a fixed $\lambda$. As proved in Parambath et al. (2014) and Koyejo et al. (2014), this baseline method should yield an approximately optimal $F_1$ score (although at a cost considerably higher than Algorithm 1 because we have to optimize $\theta$ for every $\lambda$). In our case, the baseline method is 8 times slower than Algorithm 1. To our knowledge, the baseline method is the state-of-the-art method in term of resultant $F_1$ score (not in term of efficiency). We apply weight decay to both methods and find that in general weight decay improves the performance of Algorithm 1 while hurts the performance of baseline method. For the Covertype dataset, Algorithm 1 cannot converge without weight decay, which is an evidence that weight decay may improve gradient estimation. We report results in Table 1, where "$F_1$ GS" refers to the $F_1$ score attained by the grid-search method and "$F_1$ EUM" refers to the $F_1$ score attained by Algorithm 1.

## 6 CONCLUSION

We propose an empirical utility maximization scheme that enables efficient gradient-based learning for a class of non-decomposable and non-differentiable classification performance metrics. We inquire into the proposed scheme mathematically and present preliminary experiments that validate our approach. We leave it as future work to experiment on deeper neural networks, larger datasets, and more complex performance metrics.

---

[1] https://archive.ics.uci.edu/ml/datasets/letter+recognition

[2] https://archive.ics.uci.edu/ml/datasets/adult

[3] https://archive.ics.uci.edu/ml/datasets/covertype

[4] http://www.kdd.org/kdd-cup/view/kdd-cup-2008/

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

PROOFS

**Theorem 1.** *If $F$ is a well-behaved performance metric and $C_D$ is a strongly consistent estimator of $C$, i.e.*

$$\mathbb{P}\left(\lim_{|D|\to\infty} (C_D)_{ij} = C_{ij}\right) = 1 \qquad 0 \le i, j \le |\mathcal{Y}| - 1$$

*then $F(C_D)$ is a strongly consistent estimator of $F(C)$, i.e.*

$$\mathbb{P}\left(\lim_{|D|\to\infty} F(C_D) = F(C)\right) = 1$$

*Proof.* Let $N = |\mathcal{Y}| \times |\mathcal{Y}|$. Instead of treating $C_D$ and $C$ as $|\mathcal{Y}| \times |\mathcal{Y}|$ matrices, we treat them as $N$-dimensional vectors $(C_1^D, ..., C_N^D)$ and $(C_1, ..., C_N)$. For $i = 1, ..., N$, let $E_i$ be the event that

$$\lim_{|D|\to\infty} C_i^D = C_i$$

Let $E$ be the event that

$$\lim_{|D|\to\infty} F(C_D) = F(C)$$

By the continuity of $F$,

$$\left(\forall i = 1, ..., N, \lim_{|D|\to\infty} C_i^D = C_i\right) \Rightarrow \left(\lim_{|D|\to\infty} F(C_D) = F(C)\right)$$

which implies that

$$\bigcap_{i=1}^{N} E_i \subset E$$

Taking complement on both sides, we have

$$E^c \subset \left(\bigcap_{i=1}^{N} E_i\right)^c = \bigcup_{i=1}^{N} E_i^c$$

By the monotonicity of probability measure and the union bound,

$$\mathbb{P}(E^c) \le \mathbb{P}\left(\bigcup_{i=1}^{N} E_i^c\right) \le \sum_{i=1}^{N} \mathbb{P}(E_i^c) = \sum_{i=1}^{N} 1 - \mathbb{P}(E_i) = \sum_{i=1}^{N} 1 - 1 = 0$$

Consequently,

$$\mathbb{P}\left(\lim_{|D|\to\infty} F(C_D) = F(C)\right) = \mathbb{P}(E) = 1 - \mathbb{P}(E^c) = 1 - 0 = 1$$

$\square$

**Theorem 2.** *If $F$ is a well behaved performance metric, then $\nabla_\theta F(\hat{C}_B(h_\theta))$ is a strongly consistent estimator of $\nabla_\theta F(\bar{C}(h_\theta))$, where $B \subset D$ is a mini-batch. More precisely,*

$$\mathbb{P}\left(\lim_{|B|\to\infty} \nabla_\theta F\left(\hat{C}_B(h_\theta)\right) = \nabla_\theta F\left(\bar{C}(h_\theta)\right)\right) = 1$$

*Proof.* Let $N = |\mathcal{Y}| \times |\mathcal{Y}|$. As in the proof of Theorem 1, instead of treating $\hat{C}_B(h_\theta)$ and $\bar{C}(h_\theta)$ as $|\mathcal{Y}| \times |\mathcal{Y}|$ matrices, we treat them as $N$-dimensional vectors $\hat{C}_B = (\hat{C}_1^B, ..., \hat{C}_N^B)$ and $\bar{C} =$

$(\bar{C}_1, ..., \bar{C}_N)$, where we omit the dependence on $h_\theta$ for brevity. For $i = 1, ..., N$, let $E_i$ be the event that

$$\lim_{|B| \to \infty} \nabla_\theta \hat{C}_i^B = \nabla_\theta \bar{C}_i$$

and $E$ denote the event that

$$\lim_{|B| \to \infty} \nabla_\theta F\left(\hat{C}_B\right) = \nabla_\theta F\left(\bar{C}\right)$$

Because $\nabla_\theta F(\hat{C}_i^B)$ is an unbiased estimator of $\nabla_\theta F(\bar{C}_i)$, i.e. $\mathbb{E}[\nabla_\theta F(\hat{C}_i^B)] = \nabla_\theta F(\bar{C}_i)$,

$$\mathbb{P}\left(E_i\right) = \mathbb{P}\left(\lim_{|B| \to \infty} \nabla_\theta F\left(\hat{C}_i^B\right) = \nabla_\theta F\left(\bar{C}_i\right)\right) = 1$$

by the law of large number. Consequently, suppose

$$\bigcap_{i=1}^{N} E_i \subset E$$

which is equivalent to the proposition that

$$\left(\forall i = 1, ..., N, \lim_{|B| \to \infty} \nabla_\theta \hat{C}_i^B = \nabla_\theta \bar{C}_i\right) \Rightarrow \left(\lim_{|B| \to \infty} \nabla_\theta F\left(\hat{C}_B\right) = \nabla_\theta F\left(\bar{C}\right)\right)$$

then this theorem will follow from a union bound argument similar to that in the proof of theorem 1. We now prove this proposition. Let $\hat{J}_B$ and $\bar{J}$ be the Jacobian of $\hat{C}_B$ and $\bar{C}$ w.r.t. $\theta$, i.e.

$$\hat{J}_B = \begin{bmatrix} \nabla_\theta \hat{C}_1^B \\ ... \\ \nabla_\theta \hat{C}_N^B \end{bmatrix} \qquad \bar{J} = \begin{bmatrix} \nabla_\theta \bar{C}_1 \\ ... \\ \nabla_\theta \bar{C}_N \end{bmatrix}$$

By the chain rule,

$$\nabla_\theta F\left(\hat{C}_B\right) = \nabla F\left(\hat{C}_B\right) \hat{J}_B$$

where $\nabla F\left(\hat{C}_B\right)$ is the gradient of $F$ at $\hat{C}_B$. Because $F$ is continuously differentiable, i.e. $\nabla F$ exists and is continuous,

$$\left(\forall i = 1, ..., N, \lim_{|B| \to \infty} \hat{C}_i^B = \bar{C}_i\right) \Rightarrow \left(\lim_{|B| \to \infty} \nabla F\left(\hat{C}_B\right) = \nabla F\left(\bar{C}\right)\right)$$

Thus for all $\delta > 0$, there exists $N_{F,\delta}$ such that when $|B| > N_F$,

$$|\epsilon| := \left|\nabla F\left(\hat{C}_B\right) - \nabla F\left(\bar{C}\right)\right| < \frac{\delta}{2||\bar{J}||}$$

where $||\bar{J}||$ is the matrix norm of $\bar{J}$, defined as

$$||\bar{J}|| = \sup_{|\boldsymbol{x}| \le 1} |\bar{J}\boldsymbol{x}|$$

Thus

$$\left(\forall i = 1, ..., N, \lim_{|D| \to \infty} \nabla_\theta \hat{C}_i^B = \nabla_\theta \bar{C}\right) \Rightarrow \left(\lim_{|B| \to \infty} \hat{J}_B = \bar{J}\right)$$

where the convergence is in the Frobenius norm. Because convergence in the Frobenius norm is equivalent to convergence in matrix norm, for all $\delta > 0$, there exists $N_{\hat{\jmath}}$ such that when $|D| > N_{\hat{\jmath}}$,

$$||\mathcal{E}|| := \left|\left|J_D(\theta) - J(\theta)\right|\right| < \frac{\delta}{2M}$$

where

$$M := \sup_{\boldsymbol{x} \in K} \nabla F(\boldsymbol{x}) < \infty$$

because $K$ is compact and $\nabla F$ is continuous on $K$ by definition.

Thus for all $B$ such that $|B| > \max\left\{N_F, N_{\hat{\boldsymbol{J}}}\right\}$,

$$
\begin{aligned}
\left|\nabla_\theta F\left(\hat{C}_B\right) - \nabla_\theta F\left(\bar{C}\right)\right| &= \left|\nabla F\left(\hat{C}_B\right)\hat{\boldsymbol{J}}_B - \nabla F\left(\bar{C}\right)\bar{\boldsymbol{J}}\right| \\
&= \left|\left(\nabla F\left(\bar{C}\right) + \boldsymbol{\epsilon}\right)\left(\bar{\boldsymbol{J}} + \mathcal{E}\right) - \nabla F\left(\bar{C}\right)\bar{\boldsymbol{J}}\right| \\
&= \left|\nabla F\left(\bar{C}\right)\bar{\boldsymbol{J}} + \nabla F\left(\bar{C}\right)\mathcal{E} + \boldsymbol{\epsilon}\bar{\boldsymbol{J}} + \boldsymbol{\epsilon}\mathcal{E} - \nabla F\left(\bar{C}\right)\bar{\boldsymbol{J}}\right| \\
&= \left|\nabla F\left(\bar{C}\right)\mathcal{E} + \boldsymbol{\epsilon}\bar{\boldsymbol{J}} + \boldsymbol{\epsilon}\mathcal{E} - \nabla F\left(\bar{C}\right)\bar{\boldsymbol{J}}\right| \\
&\leq \left|\nabla F\left(\bar{C}\right)\mathcal{E}\right| + \left|\boldsymbol{\epsilon}\bar{\boldsymbol{J}}\right| + \left|\boldsymbol{\epsilon}\mathcal{E}\right| \\
&\approx \left|\nabla F\left(\bar{C}\right)\mathcal{E}\right| + \left|\boldsymbol{\epsilon}\bar{\boldsymbol{J}}\right| \\
&\leq \left|\nabla F\left(\bar{C}\right)\right|\left|\left|\mathcal{E}\right|\right| + \left|\boldsymbol{\epsilon}\right|\left|\left|\bar{\boldsymbol{J}}\right|\right| \\
&\leq \frac{\delta}{2} + \frac{\delta}{2} \\
&= \delta
\end{aligned}
$$

where we ignore the high-order term $|\boldsymbol{\epsilon}\mathcal{E}|$. Consequently,

$$
\lim_{|D|\to\infty} \nabla_\theta F\left(\hat{C}_B\right) = \nabla_\theta F\left(\bar{C}\right)
$$

$\square$

**Theorem 3** (Generalization). *For a well behaved performance metric $F$, for all $\epsilon > 0$,*

$$
\lim_{|D|\to\infty} \mathbb{P}\left(\left|F(\hat{C}_D(h_\theta)) - F(\bar{C}(h_\theta))\right| < \epsilon\right) = 1
$$

*Proof.* By Theorem 1,

$$
\mathbb{P}\left(\lim_{|D|\to\infty}\left|F(\hat{C}_D(h_\theta)) - F(\bar{C}(h_\theta))\right| < \epsilon\right) = 1
$$

which implies that

$$
\lim_{|D|\to\infty} \mathbb{P}\left(\left|F(\hat{C}_D(h_\theta)) - F(\bar{C}(h_\theta))\right| < \epsilon\right) = 1
$$

$\square$

**Theorem 4** (Consistency). *For a well behaved performance metric $F$, with appropriate constraints on the capacity of parametric model $p_\theta$ (see the proof for details), we have that for all $\epsilon > 0$,*

$$
\lim_{|D|\to\infty} \mathbb{P}\left(\left|\arg\max_\theta F\left(\hat{C}_D(h_\theta)\right) - \arg\max_\theta F\left(\bar{C}(h_\theta)\right)\right| < \epsilon\right) = 1
$$

*i.e. Algorithm 2 is consistent.*

*Proof.* To prove consistency, it suffices to prove that (Vapnik (1992)):

$$
\lim_{|D|\to\infty} \mathbb{P}\left(\sup_\theta\left|F\left(\hat{C}_D(h_\theta)\right) - F\left(\bar{C}(h_\theta)\right)\right| < \epsilon\right) = 1
$$

Because $F$ is well behaved, by the union bound argument in previous proofs, it suffices to show that

$$
\lim_{|D|\to\infty} \mathbb{P}\left(\sup_\theta\left|\left(\hat{C}_D(h_\theta)\right)_{ij} - \bar{C}_{ij}(h_\theta)\right| < \epsilon\right) = 1 \qquad 1 \leq i, j \leq |\mathcal{Y}|
$$

Because $(\hat{C}_D(h_\theta))_{ij} = \frac{p_j}{n_j} \sum_{k=1}^{n_j} p_\theta(i|\boldsymbol{x}_k^j)$, it suffices to show that

$$\lim_{|D|\to\infty} \mathbb{P}\left(\sup_\theta \left|\frac{p_j}{n_j}\sum_{k=1}^{n_j} p_\theta(i|\boldsymbol{x}_k^j) - \bar{C}_{ij}\left(h_\theta\right)\right| < \epsilon\right) = 1 \qquad 1 \le i,j \le |\mathcal{Y}|$$

which holds for $p_\theta$ with finite VC-dimension by Lemma 29.1 in Devroye (2010) because

$$\mathbb{E}_D\left[\frac{p_j}{n_j}\sum_{k=1}^{n_j} p_\theta(i|\boldsymbol{x}_k^j)\right] = \bar{C}_{ij}\left(h_\theta\right)$$

$\square$

## WELL BEHAVED PERFORMANCE METRICS

The following is a non-exhaustive list of non-decomposable and non-differentiable, yet well-behaved performance metrics in the setting of binary classification. They can be extended to the setting of multi-class classification in the same way that the $F$-measure is extended to multi-class classification in Section 3.

- $\text{AUC} = \frac{\text{fp} \cdot \text{fn}}{(\text{tp}+\text{fn})(\text{fp}+\text{tn})}$
- $F_\beta = (1 + \beta^2) \cdot \frac{p^+ - \text{fn}}{(1+\beta^2)p^+ - \text{fn} + \text{fp}}$
- $\text{G-Mean} = \sqrt{\text{tp} \cdot \text{tn}}$
- $\text{Jaccard} = \frac{\text{tp}}{\text{tp}+\text{fp}+\text{fn}}$
- $\text{Q-Mean} = 1 - \sqrt{\frac{(1-\text{tp})^2 + (1-\text{tn})^2}{2}}$

We refer interested readers to Choi et al. (2010) for a more exahustive list of these performance metrics.

