# OpenReview forum: "Gradient-based learning for F-measure and other performance metrics"
_ICLR.cc/2019/Conference_

### Official Review · AnonReviewer2 · 2018-10-27
**Marginally below acceptance**

**Rating:** 5
**Confidence:** 5

**Review:**

This paper studies the problem of optimizing non-decomposable metric in classification. This topic has been discussed in several recent works mainly under deterministic classifier context, the authors discuss the possibility of training a neural network and learn the model by gradient-based methods, which could result in randomized classifier; and conducted experiments to compare the performance with other existing methodologies. I have the following concerns after reading it.

1.The main idea of the paper has shown in other related works and the authors didn’t convince me why their work solves something that could not be solved in existing work. The related work section missed some relevant recent work including Ref[1], in which the method is also gradient-based and can be applied to neural networks. The well-behaved notion used in Definition 2 seems much weaker than the assumptions shown in Ref[1,2] to guarantee existence or uniqueness of the Bayes classifier, the authors could spend some effort to discuss why they require less assumptions.

2.For the theory part, all the convergence results are proved in an asymptotic way without further discussion in the sample complexity. This becomes problematic for this work because (as shown in eq (7)) mini batch size goes to infinity is an unrealistic assumption in neural network training. Also when the class is unbalanced, empirical mean converging to population also slows down significantly which is required in Eq (4) and other places. I would like to see more discussion on the sample complexity either theoretically or experimentally.

3.The experiments lack details for reproducing the results or generalizing the gain to other problems. For example, batch size, learning rate or how the size of the network influence the performance metrics. This information will be useful for others who want to apply the proposed method.

There are some minor formatting issues like the leading space in \citep. Please fix those.

Based on the above reasons, I’ll give this paper a 5.

[Ref 1] Yan, B., Koyejo, S., Zhong, K. & Ravikumar, P.. (2018). Binary Classification with Karmic, Threshold-Quasi-Concave Metrics. Proceedings of the 35th International Conference on Machine Learning, in PMLR 80:5531-5540
[Ref 2] Narasimhan, H., Kar, P., & Jain, P. (2015, June). Optimizing non-decomposable performance measures: a tale of two classes. In International Conference on Machine Learning (pp. 199-208).

---

> ### Author Response · Authors · 2018-11-14
> **Responses**
>
> Dear Reviewer,
>
> First, thank you for your helpful review!
>
> Regarding issues you mentioned:
>
> 1. The idea of optimizing non-decomposable objectives by leveraging their mathematical properties is indeed not new. However, we believe that the formulation and analysis of the proposed surrogate objective is novel. [Ref 1] is very interesting and we will consider it as a baseline for our method. We did not consider the existence and uniqueness of the Bayes classifier. We are only interested in whether our surrogate objective can yield best-in-class classifiers given a fixed hypothesis class (Theorem 4), which is why our results only require a weaker condition.
>
> 2. Yes, we should discuss sample complexity, especially when discussing the impact of batch size. The convergence of empirical mean to population indeed can become a problem when there are very few samples for a class. However, this issue is less severe for large datasets, where minority classes have numerous samples despite their low frequencies.
>
> 3. We will surely include more details about experiments. We will also release our code soon.

---

### Official Review · AnonReviewer1 · 2018-10-29
**Needs more theoretical or experimental support.**

**Rating:** 3
**Confidence:** 4

**Review:**

Update: I still feel that the paper should have either strong theory, strong experiments, or some of each to be accepted, but that both are lacking. The revisions required would be too great for acceptance at this time.

Original review:
The paper proposes a general method to optimize for performance metrics which can be written in terms of the entries of the confusion matrix. The idea is to approximate the entries of the confusion matrix using their expected values for a randomized classifier, plug these estimates into the formula for the desired metric, and optimize that quantity. This is a compelling idea but it needs more support than the theoretical or experimental sections give.

The simplicity and generality of the method are appealing. Smooth surrogates derived from randomized classifiers have been considered in the context of accuracy [1] and other performance measures [2, 3] and the paper should include some discussion of this prior work, but to my knowledge the broad applicability to non-decomposable and non-differentiable metrics expressible in terms of the confusion matrix is new.

The theoretical sections could use some improvement. It is worth mentioning that the loss obtained with the proposed method is nonconvex. The first equation in theorem 1 is described with “... where convergence in probability is entry-wise”, when the equation refers to almost sure convergence for a scalar, not convergence in probability for entries of a matrix.

No convergence rates are given, only asymptotic almost sure convergence as the size of the dataset or the minibatch goes to infinity. For finite datasets these statements are obvious, and while convergence is reassuring for infinite datasets, I imagine the rates will look very different for the loss (a scalar) and the gradient (which may have millions of coordinates). Theorem 3 considers the generalization of a single classifier which is independent of the empirical sample, which makes it irrelevant to cases where the model is learned. Theorem 4, which seeks to give a uniform bound over the model class, only shows that generalization occurs in the limit of infinitely much data (which is not surprising or particularly interesting).

The experimental section compares the algorithm against a well-known and strong baseline, but without any information about the variance of the results and only for a deep network. Several questions remain: Where the proposed method improves over the baseline, is this improvement due to the new method or the interaction between the method and the model? How would the method perform on e.g. a linear model, which is better understood? How do the results depend on batch size, which affects the bias in the gradients?

[1] Roux, Nicolas Le. "Tighter bounds lead to improved classifiers." arXiv preprint arXiv:1606.09202 (2016).
[2] Mozer, Michael C., et al. "Prodding the ROC curve: Constrained optimization of classifier performance." Advances in Neural Information Processing Systems. 2002.
[3] Goh, Gabriel, et al. "Satisfying real-world goals with dataset constraints." Advances in Neural Information Processing Systems. 2016.

---

> ### Author Response · Authors · 2018-11-14
> **Responses**
>
> Dear Reviewer,
>
> First, thank you for your helpful review!
>
> We agree that we should provide convergence rates. For the convergence of scalars (e.g., in Theorem 4), it would not be hard to derive convergence rates by replacing the law of large number in our analysis with concentration inequalities. The convergence rates of gradients, as you mentioned, is trickier and we will further study this. Also, replacing Theorem 3 with a data-dependent generalization bound may require more efforts and we will further study this.
>
> We believe that the convexity of the proposed surrogate objective depends on the convexity of the performance metric that we are concerned with. With a convex performance metric, the proposed performance metric is convex. For example, in the case where the performance metric is accuracy, the proposed surrogate objective is simply a sum of posterior probabilities inferred by our classifier. Of course, the objective cannot be convex with a non-convex performance metric.
>
> The first equation in Theorem 1 will be fixed.
>
> We will augment the experimental section to provide more information. Works you mentioned are very helpful. We will include them in our updates.

---

> > ### Comment · AnonReviewer1 · 2018-11-26
> > **nonconvexity**
> >
> > Even for accuracy, the expected classification error results in a nonconvex loss. See the Le Roux reference I mentioned.

---

> > > ### Author Response · Authors · 2018-11-26
> > > **nonconvexity**
> > >
> > > Yes, the loss is nonconvex w.r.t. \theta, even in the case of accuracy. (I was thinking about convexity w.r.t. the classifier's posterior probabilities when writing the response)

---

### Official Review · AnonReviewer3 · 2018-11-05
**An interesting paper with some issues**

**Rating:** 5
**Confidence:** 3

**Review:**

This paper proposes a gradient-based learning for F1 measure under the utility maximization framework. F1 is a widely used evaluation metric in information retrieval and machine learning, and it is hard to optimize as it is non-decomposable and non-differentiable. This research direction is hence extremely interesting.

The paper is well organized and easy to follow. The general methodology seems sound. Below are some detailed comments.

- Page 1, Section 2.1. The notation of the probabilistic classifier is not typed correctly.

- Page 7. The result strongly depends on how well Eq. (5) holds. Two critical assumptions regarding the data are made here, (1) D -> ∞ and (2) B -> ∞. The first assumption is implicitly confirmed in the experiments, as in Table 1 the proposed method outperforms when the sample size is big. I am a little bit puzzled about the second assumption though. Eq. (7) holds, (and consequently Eq. (5)) when B -> ∞, but it cannot be the case in practice, since B tends to have moderate sizes. I wonder how this impacts the results. Batch size isn't discussed at all in the experiments. The discussion on noise control is nice, but it doesn't contribute to the validation of Eq. (5) or Eq. (7).

- Algorithm 1 & 2. It may be a good idea to be explicit what the outputs of the algorithms are. The algorithms are referenced by their section numbers instead of their algorithm numbers.

- The experimental section can be extended. The paper has extensively discussed other well-behaved metrics and tasks beyond binary classification. None of these are tested empirically.

- If I understand correctly, the GS method, with a much higher computational cost, is near optimal. If so, its results should serve as an empirical upper-bound for F1. Then how come the proposed method outperforms it on 4 over 6 dataset?

- There are additional references on F1 maximization. To name a few: (1) Chai. Expectation of F-measures. SIGIR 2005. (2) Waegeman et al. On the Bayes-Optimality of F-Measure Maximizers. JMLR.

---

> ### Author Response · Authors · 2018-11-14
> **Responses**
>
> Dear Reviewer,
>
> First, thank you for your helpful review!
>
> Regarding issues you mentioned:
>
> - Page 1, Section 2.1. Could you please specify where the typo is?
>
> - Page 7. Indeed, when analyzing the impact of batch size, we should have given the rate of convergence instead of only considering asymptotic behaviors.
>
> - Algorithm 1 & 2. Fixed.
>
> - We will include an experiment with multi-class F1 score, in which case the GS method is proved optimal as well. For other performance metrics, we are still looking for strong baselines.
>
> - We suspect that the GS method is outperformed because we did not use sufficiently dense grids when grid-searching \lambda. The reason for not using denser grids is that we also have to grid-search other hyper-parameters, e.g. learning rate, and thus using denser grids for the GS method will significantly increase computation time. Using the step size mentioned in our paper, we need almost two days to finish all experiments with the GS method on a 4-GPU workstation. Thus halving the step size in our case will cost almost another two days.
>
> - We did not notice references you mentioned and will include them in our paper.

---

### Author Response · Authors · 2018-11-06
**Revision**

Dear Reviewers and Readers,

We notice that there are typos in our submission. Also, there are places where our presentation are not very clear. We have therefore uploaded a revision of our submission.

We apologize for any confusion caused by the original manuscript.

Best,
Authors

---

### Meta-Review · Area_Chair1 · 2018-12-16

**Confidence:** 5
**Recommendation:** Reject

**Metareview:**

This manuscript proposes a gradient-based learning scheme for non-differentiable and non-decomposable metrics. The key idea is to optimize a soft predictor directly (instead of aiming for a deterministic predictor), which results in a differentiable loss for many of these metrics. Theoretical results are provided which describe the performance of this approach.

The reviewers and ACs noted weakness in the original submission related to the clarity of the presentation and novelty as related to already published work. There was also a concern about the usefulness the main theoretical results due to asymptotic assumptions. The manuscript would be significantly strengthened if the reliance on infinite sample sizes is resolved, or sufficient empirical evidence is provided which suggests that the asymptotic issues are not practically significant.